# Somatic chromosomal engineering identifies BCAN-NTRK1 as a potent glioma driver and therapeutic target

Peter J. Cook[1], Rozario Thomas[1,2], Ram Kannan[1], Esther Sanchez de Leon[2], Alexander Drilon[3], Marc K. Rosenblum[4], Maurizio Scaltriti[4,5], Robert Benezra[1] & Andrea Ventura[1]

The widespread application of high-throughput sequencing methods is resulting in the identification of a rapidly growing number of novel gene fusions caused by tumour-specific chromosomal rearrangements, whose oncogenic potential remains unknown. Here we describe a strategy that builds upon recent advances in genome editing and combines *ex vivo* and *in vivo* chromosomal engineering to rapidly and effectively interrogate the oncogenic potential of genomic rearrangements identified in human brain cancers. We show that one such rearrangement, an microdeletion resulting in a fusion between Brevican (BCAN) and Neurotrophic Receptor Tyrosine Kinase 1 (NTRK1), is a potent oncogenic driver of high-grade gliomas and confers sensitivity to the experimental TRK inhibitor entrectinib. This work demonstrates that BCAN-NTRK1 is a *bona fide* human glioma driver and describes a general strategy to define the oncogenic potential of novel glioma-associated genomic rearrangements and to generate accurate preclinical models of this lethal human cancer.

[1] Cancer Biology and Genetics Program, Memorial Sloan Kettering Cancer Center, New York, New York 10065, USA. [2] Department of Molecular Biology, Weill Cornell Graduate School of Medical Sciences of Cornell University, New York, New York 10065, USA. [3] Thoracic Oncology Service, Department of Medicine, Memorial Sloan Kettering Cancer Center, New York, New York 10065, USA. [4] Department of Pathology, Memorial Sloan Kettering Cancer Center, New York, New York 10065, USA. [5] Human Oncology and Pathogenesis Program, Memorial Sloan Kettering Cancer Center, New York, New York 10065, USA. Correspondence and requests for materials should be addressed to R.B. (email: benezrar@mskcc.org) or to A.V. (email: venturaa@mskcc.org).

Ongoing large-scale sequencing efforts provide an unprecedented picture of the genetic complexity of human cancers. Deciphering this wealth of data and separating true driver mutations from benign passengers is essential to better understand cancer pathogenesis and develop more effective targeted therapies. Mouse models of human cancers have proven invaluable in defining the oncogenic potential of specific genomic lesions. However, the generation of new mouse models via conventional gene targeting methods is time-consuming and costly, and cannot be scaled up to investigate the functional relevance of the myriad recurrent cancer-associated mutations that have been—and are being—identified in patient samples.

Among these mutations, chromosomal rearrangements are of particular interest as they can result in the generation of therapeutically actionable gene fusions[1–3]. The importance of this class of cancer-associated mutations has become even more evident in recent years, as unbiased approaches involving deep-sequencing technology have led to a dramatic increase in the number of reported gene fusions in cancer[4,5]. Determining which among the thousands of newly identified gene fusions are oncogenic drivers and confer sensitivity to targeted therapies remains a crucial and largely unmet challenge[6].

CRISPR-based somatic genome editing provides an appealing alternative to genetic engineering in embryonic stem cells[7]. For example, we and others have demonstrated that the simultaneous expression of Cas9 and two guideRNAs (gRNAs) targeting the desired breakpoints can be used to induce specific chromosomal rearrangements in cultured cells and directly in somatic cells of adult mice[8–12].

We previously demonstrated that infecting the lungs of adult mice with recombinant adenoviral vectors expressing Cas9 and two guide RNAs targeting the relevant intronic regions on Chromosome 17 is sufficient to generate the *Eml4-Alk* chromosomal inversion and promote the formation of lung adenocarcinomas that closely recapitulate the histological and biological features of human *EML4-ALK*-driven lung cancers[10].

These initial 'proof of concept' studies focused on modelling a well-known oncogenic driver of non-small cell lung cancer for which effective targeted therapy was already available. In the present study, we use a combination of *ex vivo* and *in vivo* somatic chromosomal engineering to explore the oncogenic potential, and the therapeutic vulnerability, of largely uncharacterized chromosomal rearrangements in brain cancer. Our choice to focus on primary malignant gliomas was motivated by the paucity of therapeutic options available to patients affected by these cancers[13] and by the high volume of genetic data available from human gliomas[14–16]. Glioblastoma, the most common type of brain cancer in adults, was the first tumour type analysed as part of the cancer genome atlas (TCGA) project[16]. This and additional sequencing efforts have revealed the broad spectrum of genomic alterations[14,15], as well as the high degree of intertumoral[17] and intratumoral[18] heterogeneity that characterize this tumour type. Constitutive activation of receptor tyrosine kinase signalling is a major feature of these tumours, as illustrated by the frequent amplification of the *EGFR* locus. Other common genetic lesions include *TERT* promoter mutations, loss of *PTEN* or *TP53*, and mutation of *IDH1* and *NF1* (refs 14,16,19). Although these common mutations have been the subject of extensive studies, significantly less is known about the more rare genetic events, including many chromosomal rearrangements, which are found in only a small subset of patients but could be effective therapeutic targets.

In the present study we used CRISPR-Cas9 genome editing *ex vivo* and *in vivo* to test the oncogenic potential of the chromosomal rearrangements underlying four glioma-associated gene fusions. We show that the chromosomal deletion generating the *Bcan-Ntrk1* gene fusion can drive the formation of high-grade gliomas and confers sensitivity to entrectinib, an experimental TrkA inhibitor. These results demonstrate the feasibility of CRISPR-Cas9 genome editing as a tool to rapidly generate preclinical models of brain cancer in mice.

## Results

**In vitro generation of glioma-associated gene fusions**. We examined public databases and the scientific literature to identify poorly characterized gene fusions found in human gliomas[5,20]. We focused on fusions involving 'druggable' kinases with exonic structure and relative gene orientation conserved between mouse and human. While multiple gene fusions met these criteria, we selected four for the present study: *FGFR3-TACC3*, *SEC61G-EGFR*, *BCAN-NTRK1* and *GGA2-PRKCB*. The first three involve receptor tyrosine kinases and have been reported in high-grade gliomas, while *GGA2-PRKCB* involves a serine/threonine protein kinase C family member and has been found in low-grade gliomas[5,20].

Of the four gene fusions, *FGFR3-TACC3*—resulting from a tandem duplication on human chromosome 4—is the best characterized, with an estimated incidence of $\sim 3\%$ (ref. 21). Fusions involving the tyrosine kinase domain of *EGFR* have been identified in 3.7% of glioblastoma patients[22], while fusions involving *NTRK1*, which encodes the TrkA tyrosine kinase receptor, are collectively found in 1–2.5% of glioblastomas[23,24]. The four rearrangements include one intrachromosomal deletion, two inversions and one tandem duplication (Fig. 1a and Supplementary Figs 1–3). In each case the entire region involved in the rearrangement is syntenic between mouse and human. Importantly, none of them has been previously modelled in mice, and only the FGFR3-TACC3 fusion protein has been previously validated as a potential tumour driver based on viral expression of its cDNA in the mouse brain[21].

We first attempted to model these rearrangements in primary adult neural stem cells (aNSCs)[25] isolated from the brain of $p53^{-/-}$ mice. This genetic background was selected because inactivation of the p53 pathway is a common event (85.3%) in human glioblastomas[14]. To generate the rearrangements we designed gRNA pairs targeting the desired intronic breakpoints and expressed them, together with Cas9 and a GFP marker, in aNSCs via plasmid nucleofection (Fig. 1b and Supplementary Table 1). Generation of the predicted chromosomal rearrangements was first tested by PCR on genomic DNA, and then confirmed by sequencing (Fig. 1c and Supplementary Figs 1–3). In each case, we also confirmed expression of the corresponding fusion transcript by reverse transcription (RT)–PCR and direct sequencing (Fig. 1d and Supplementary Figs 1–3). As expected, and as previously described[10], expression of each gRNA pair not only produced the desired rearrangement but also, in a subset of cells, caused the formation of additional rearrangements. For example, in cells transfected with the gRNA pair designed to induce the *Bcan-Ntrk1* deletion, we also detected the PCR product diagnostic of the inversion of the same $\sim 205$ kb region (Fig. 1c).

Collectively, these results indicate that somatic genome engineering using the CRISPR/Cas system can be effectively used to induce a wide range of chromosomal rearrangements in primary aNSCs.

**The Bcan-Ntrk1 rearrangement is a glioma driver**. Having established that the four desired rearrangements can be generated in primary *p53-null* aNSCs using CRISPR/Cas-based gene editing, we next sought to test whether any of them were sufficient to

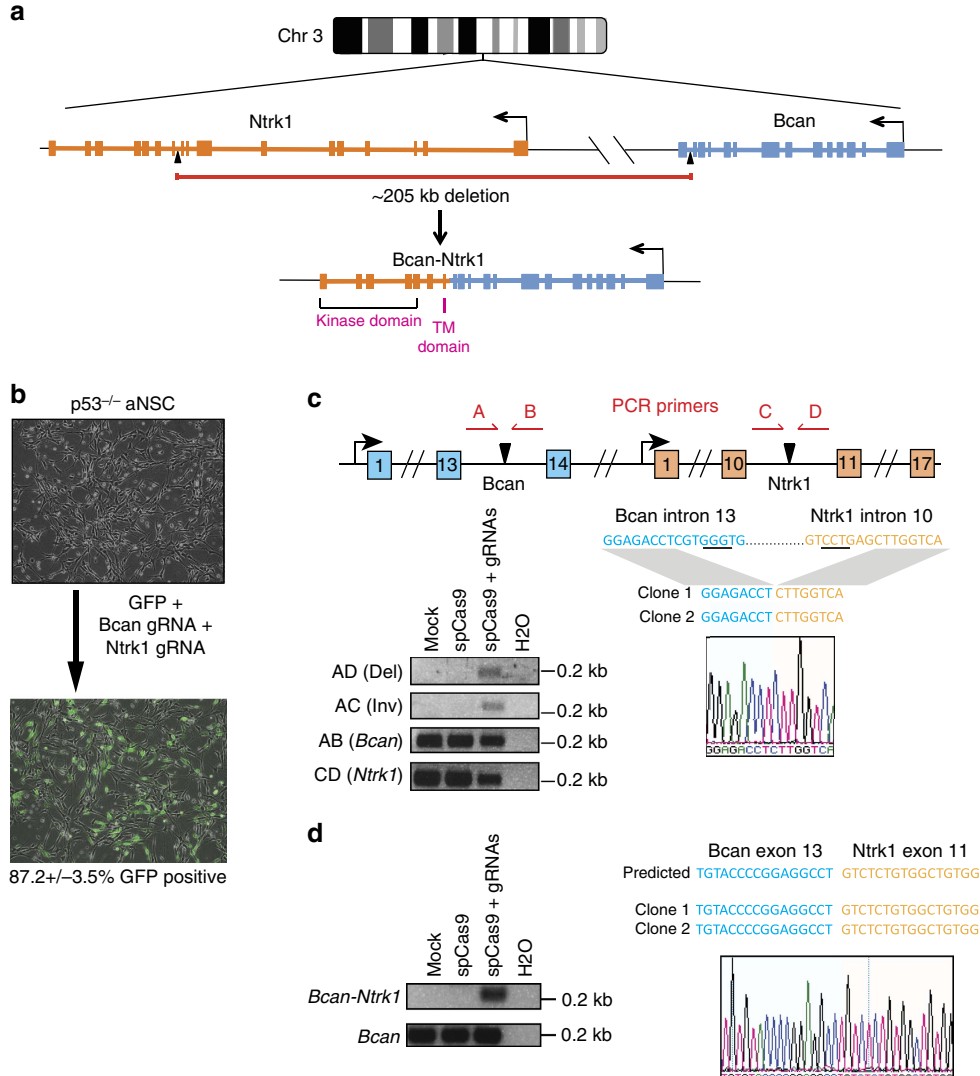

**Figure 1 | Induction of Bcan-Ntrk1 deletion in murine cells using CRISPR-Cas9.** (**a**) Schematic representation of the deletion on mouse chromosome 3 generating the *Bcan-Ntrk1* fusion allele. gRNA cut sites are indicated by black arrowheads. (**b**) Representative GFP fluorescence images of $p53^{-/-}$ mouse adult neural stem cells (aNSCs) before and 24 h after nucleofection with indicated expression plasmids. FACS analysis was used to determine the percent of GFP-positive cells (mean ± s.e.m., $n = 3$). (**c**) Top, schematic of the targeted intronic region in *Bcan* and *Ntrk1*. Arrowheads indicate cleavage sites by the gRNAs. Red arrows indicate PCR primers (**a–d**) designed to detect the rearrangement. PCRs were performed on genomic DNA extracted from aNSCs nucleofected with the indicated pX330 constructs (bottom left). The PCR bands were sub-cloned and the sequences of two independent clones and a representative chromatogram are shown in the lower right panel. (**d**) RT–PCR (left panel) on total RNA extracted from aNSCs nucleofected with the indicated constructs using primers designed to detect the wild-type *Bcan* cDNAs or the *Bcan-Ntrk1* fusion transcript. The band corresponding to the fusion transcript was subcloned and sequenced (right panel).

transform these cells. We orthotopically implanted aNSCs nucleofected with the four gRNA pairs into the brain of immunodeficient mice (NCr-*Foxn1^{nu}*)[26] and monitored the animals for up to 150 days for the development of brain tumours. None of the mice implanted with control $p53^{-/-}$ aNSCs transfected with Cas9 alone, or with Cas9 and gRNA pairs designed to induce the *Fgfr3-Tacc3*, *Sec61g-Egfr*, or *Gga2-Prkcb* rearrangements developed tumours during the observation period (Supplementary Table 1). In contrast, four out of five injected with cells nucleofected with Cas9 and the *Bcan-Ntrk1* gRNA pair (gRNA-BN1/Cas9) developed tumours at the site of implantation and had to be killed within 125 days (Fig. 2a). The tumours exhibited features of high-grade astrocytic neoplasm with little variation in morphology across all samples examined. Histologically, they displayed densely packed nuclei and distinctive areas of necrosis (Fig. 2b), and grew as

relatively compact masses of somewhat epithelioid to more conspicuously spindled cells with eosinophilic cytoplasmic processes and vesicular nuclei containing small nucleoli and multiple chromocentres (Fig. 2c). Mitotic figures were readily identifiable, and pleomorphic multinucleated tumour giant cells were a constant finding, these constituting minor elements within more monomorphous neoplastic cell populations (Fig. 2c). Ki67 staining indicated strong proliferation, and uniform expression of GFAP and Olig2 confirmed the glial nature of all tumours (Fig. 2d). Staining with the stem cell marker Nestin, which typically labels a small subpopulation of tumour cells localized near the vasculature[27], was also positive in a fraction of cells in the tumours examined (Fig. 2d). Notably, every tumour analysed ($n = 4$) was positive for the intrachromosomal deletion resulting in the *Bcan-Ntrk1* fusion allele (Fig. 2e) and expressed the *Bcan-Ntrk1* fusion transcript (Fig. 2f). In two tumours,

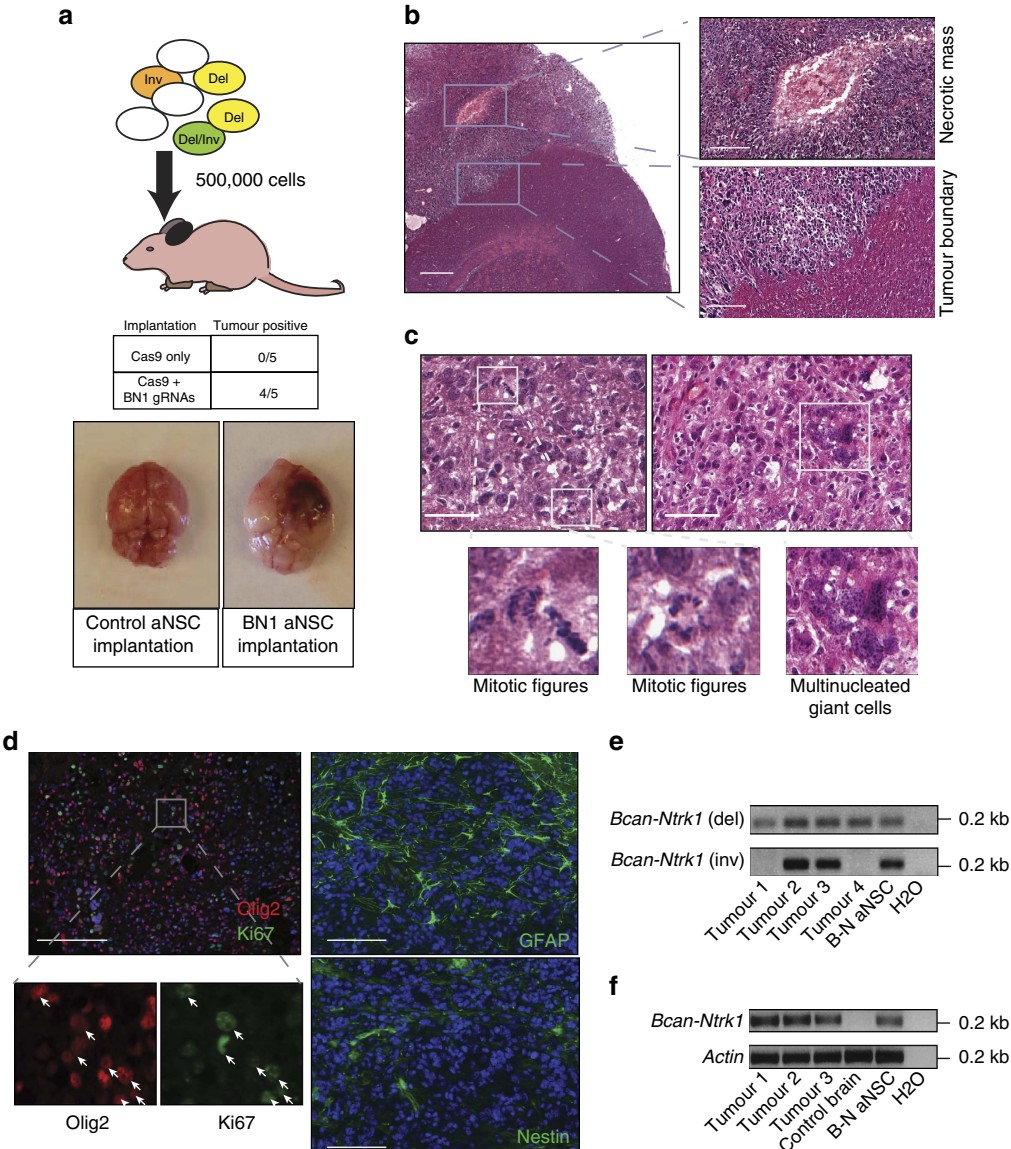

**Figure 2 | aNSCs harbouring the Bcan-Ntrk1 deletion form high-grade gliomas in mice.** (**a**) $p53^{-/-}$ aNSCs nucleofected with the BN1/Cas9 plasmids or with plasmids expressing Cas9 only were injected into the brain of nude mice (500,000 cells/mouse, top panel). Number of mice developing brain tumours is reported in the middle panel. Representative tumour-bearing and control brains are shown (bottom). (**b**) Haematoxylin and eosin staining from a representative brain tumour showing dense cellularity relative to adjacent non-tumour regions, (left panel, lower inset), and the presence of large necrotic regions (left panel, upper inset). Scale bar = 0.5 mm (left panel) or 0.2 mm (right panels). (**c**) Examples of mitotic figures (left) and multi-nucleated giant cells (right). Scale bar = 50 μm. (**d**) Immunofluorescence staining of tumours for Olig2, Ki67, GFAP and Nestin. Arrows point to cells positive for Olig2 and Ki67. Scale bar = 0.1 mm. (**e**) PCR analysis of genomic DNA from bulk tumour tissue to detect the *Bcan-Ntrk1* deletion and inversion. (**f**) RT–PCR analysis of total RNA purified from bulk tumour tissue showing expression of the *Bcan-Ntrk1* fusion transcript in each tumour.

in addition to the deletion, we also detected the corresponding chromosomal inversion (Fig. 2e). No tumours were observed when mice were implanted with wild-type aNSC cultures bearing the *Bcan-Ntrk1* deletion, indicating that concomitant inactivation of the p53 pathway is required for *Bcan-Ntrk1*-driven tumour progression (Supplementary Table 1).

To further characterize the transforming potential of the *Bcan-Ntrk1* rearrangement we derived single-cell clones from $p53^{-/-}$ aNSCs nucleofected with the gRNA-BN1/Cas9 constructs and genotyped them. Clonal analysis revealed the presence of the *Bcan-Ntrk1* deletion and inversion in 3.43% and 13.63% of clones, respectively. We selected three clones for further analysis: clone #5 harbouring only the *Bcan-Ntrk1* deletion (Fig. 3a), clone #9 harbouring only the inversion (Fig. 3b) and clone #8

harbouring both the inversion and the deletion (Supplementary Fig. 4). Additional subcloning of these lines confirmed their clonal nature (Fig. 3a,b and Supplementary Fig. 4). We then tested the tumour-forming potential of these three clones by implanting each into nude mice via orthotopic intracranial injection. The clonal lines harbouring the intra-chromosomal deletion and expressing *Bcan-Ntrk1*, clones #8 and #5, generated high-grade gliomas with full penetrance, closely resembling tumours derived from the mixed parental line (Fig. 3c and Supplementary Fig. 4b). As expected, the resulting tumours also matched the genotype of the clone of origin (Fig. 3d and Supplementary Fig. 4c). In contrast, the clone harbouring the inversion but not the deletion (clone #9) failed to produce tumours (Fig. 3e), further confirming that expression of the *Bcan-*

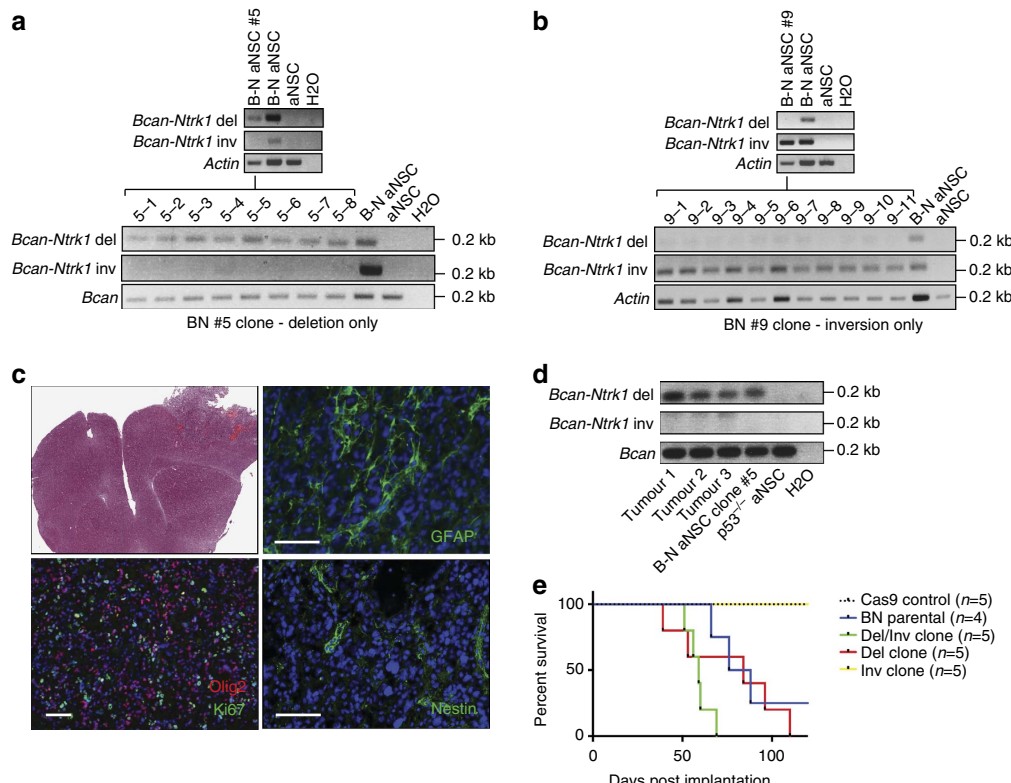

**Figure 3 | Clonal analysis demonstrates that the Bcan-Ntrk1 deletion is required for glioma formation.** (**a**,**b**) Genomic PCR analysis of single-cell-derived clones established from $p53^{-/-}$ aNSCs nucleofected with the Cas9/BN1 plasmids. Clones positive for the deletion allele (clone #5, **a**) or the inversion (clone #9, **b**) are shown. (**c**) H&E and immunofluorescence staining of tumours obtained from mice injected with clone #5. Scale bar = 0.1 mm. (**d**) Genomic PCR of tumours derived from clone #5 confirming the presence of the deletion, but not of the inversion. (**e**) Kaplan–Meier curves of mice implanted with the indicated aNSC cells.

*Ntrk1* fusion transcript generated by the chromosomal deletion is necessary for cellular transformation.

These results strongly suggest that the *Bcan-Ntrk1* deletion is oncogenic, but it is formally possible that additional off-target mutations induced by the gRNA pair could also contribute to the process. To test this possibility, we designed a second pair of gRNAs (gRNA-BN2), cutting at different sites within the same introns targeted by gRNA-BN1 (Supplementary Table 2). Expression of these gRNAs and Cas9 in $p53^{-/-}$ aNSCs also generated the *Bcan-Ntrk1* rearrangement and led to the formation of *Bcan-Ntrk1*-positive high-grade gliomas upon implantation into the brain of nude mice (Supplementary Fig. 5). Based on these results we conclude that induction of the *Bcan-Ntrk1* rearrangement in a *p53-null* background is sufficient to initiate glioma formation.

**Bcan-Ntrk1 gliomas respond to an Ntrk1 inhibitor.** Tumours that are driven by oncogenic fusion kinases frequently display strong therapeutic response to pharmacological inhibitors specific to the kinase[1–3]. To determine whether *Bcan-Ntrk1* positive brain tumours are addicted to TrkA kinase signalling, we derived cell lines from two independent gliomas arising in mice implanted with gRNA-BN1/Cas9-nucleofected aNSCs (hereafter referred to as BNN4 and BNN2). PCR genotyping of each tumour line demonstrated that BNN4 cells were positive for both the *Bcan-Ntrk1* deletion and the corresponding inversion, while BNN2 was positive only for the *Bcan-Ntrk1* deletion (Supplementary Fig. 6a). Both cell lines expressed the *Bcan-Ntrk1* fusion transcript and readily formed high-grade gliomas with short latency when orthotopically implanted into nude mice (Supplementary Fig. 6b–d).

The *Bcan-Ntrk1* fusion lacks the ligand-binding domain of full-length wild-type *Ntrk1*, and is predicted to activate downstream signalling pathways, including the MAPK and PI3K–AKT pathways, in a ligand-independent manner[28].

Treatment of these tumour lines with nanomolar concentrations of entrectinib (RXDX 101), an investigational Trk inhibitor[29,30], resulted in reduced cell viability and inhibition of Akt and Erk phosphorylation in *Bcan-Ntrk1*-positive glioma lines, but not in a control glioma line generated by ectopically expressing PDGFb (Fig. 4a,b)[31]. Interestingly, we did not observe activation of Caspase-3, suggesting that entrectinib does not induce apoptosis in *Bcan-Ntrk1*-positive glioma cells (Supplementary Fig. 7a)

To explore the therapeutic potential of entrectinib in *Bcan-Ntrk1*-driven gliomas in an *in vivo* setting, we then orthotopically injected BNN4 tumour cells into nude mice and randomly divided the animals into entrectinib or vehicle treatment groups. At day 12 post implantation, we initiated daily oral gavage with either entrectinib or vehicle[29,32], and continued the treatment for 14 days. All animals in the control group succumbed to brain tumours within the 14-day treatment window, while all animals in the entrectinib group survived during the same period (Fig. 4c, $P<0.0001$, log-rank test) and showed weight gain relative to the control group (Supplementary Fig. 7b). Gliomas initiated using the BNN2 tumour cell line showed a similar response to entrectinib treatment *in vivo* (Supplementary Fig. 7c). Histological staining performed on entrectinib-treated tumours collected immediately after the last entrectinib dose showed that while tumour tissue was clearly present, the proliferative index of this tissue was significantly reduced in comparison to control tumours with no increase in

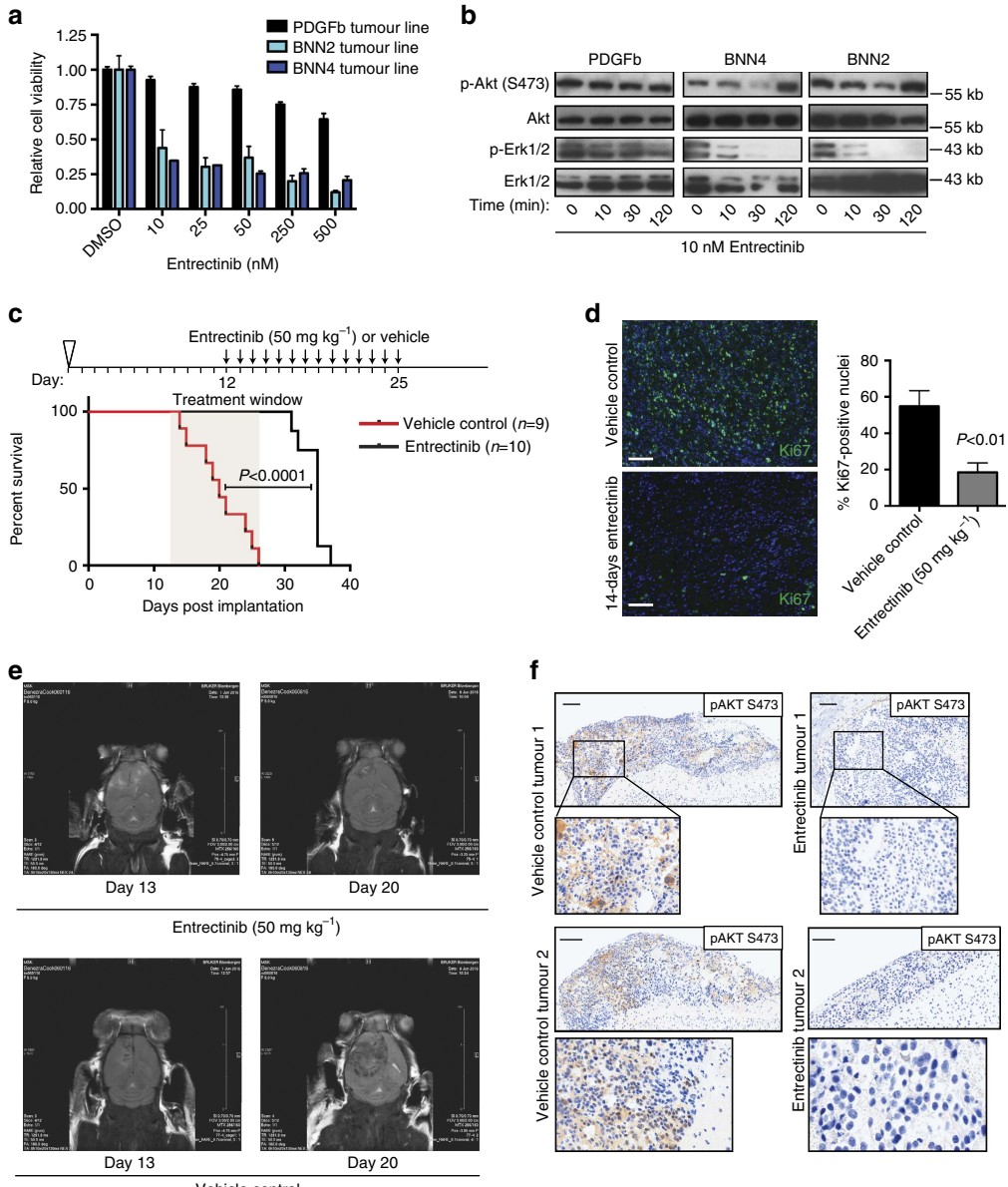

**Figure 4 | Entrectinib inhibits Bcan-Ntrk1-positive glioma growth *in vivo*.** (**a**) Cell viability assays performed using a metabolic dye (Alamar Blue) were used to assess the response of two independent Bcan-Ntrk1 positive (BNN4 and BNN2) and a control (PDGFb) glioma cell lines to 24 h treatment with entrectinib. Cells underwent 24 h growth factor starvation before treatment. Error bars represent mean ± s.e.m. (**b**) Western blot analysis of three glioma lines examining phosphorylation events downstream of Ntrk1 activation in response to 10 nM entrectinib at indicated time points. Cells underwent 24 h growth factor starvation before treatment. (**c**) Upper panel: schematic of the *in vivo* response to entrectinib experiment. Mice were injected intracranially with cells derived from a glioma harbouring the *Bcan-Ntrk1* rearrangement at day 0. Daily treatment with entrectinib or vehicle was initiated at day 12 and continued for 14 days. Lower panel: Kaplan–Meier plot of mice receiving entrectinib or vehicle (*P*-value = log-rank test). (**d**) Immunofluorescence staining for Ki67 (left panel) and quantification of Ki67 + cells (right panel) of treated and control tumours collected on day 26 post-implantation. Scale bar = 0.1 mm. *P*-value = two tailed *t*-test. Error bars represent mean ± s.e.m. (**e**) Representative magnetic resonance imaging analysis performed on entrectinib- or vehicle-treated mice on day 13 and 20 post-implantation. (**f**) Histological analysis of animals receiving either entrectinib or vehicle for 2 days (day 14 post tumour cell implantation). Immunostaining for pAkt1 (S473) within tumours is shown. Scale bar = 0.1 mm.

apoptotic cells (Fig. 4d and Supplementary Fig. 7d). Small animal magnetic resonance imaging analysis performed on a subset of animals on days 13 and 20 post tumour implantation (days 1 and 7 of treatment) showed clear tumour progression in the control-treated animals, but not in the entrectinib-treated group (Fig. 4e). Consistent with a cytostatic effect, the remaining animals not killed for tumour analysis succumbed rapidly to brain tumour after treatment with entrectinib was suspended. In a separate group of animals injected with the *Bcan-Ntrk1* tumour cells, immunostainings performed after 2 days of treatment with

entrectinib revealed a strong decrease in Akt phosphorylation at Serine 473, confirming inhibition of TrkA signalling *in vivo* (Fig. 4f). Collectively, these results indicate that entrectinib is an effective inhibitor of *Bcan-Ntrk1*-driven brain tumours.

**An autochthonous mouse model of Bcan-Ntrk1-driven glioma.** Orthotopic implantation of primary cells modified *ex vivo* is useful to rapidly assess the oncogenic potential of chromosomal rearrangements and offers the obvious advantage of allowing the

introduction of additional genetic modifications. However, *ex vivo* manipulation of aNSCs does not fully reflect the natural evolution of primary brain cancers, and it is therefore essential to demonstrate that these rearrangements can transform cells directly *in situ* in an immunocompetent host.

We therefore generated an all-in-one adenoviral vector expressing FLAG-spCas9 and the sgRNA-BN1 pair (Ad-BN; Fig. 5a) to engineer the *Bcan-Ntrk1* rearrangement directly in the brain of adult mice. When administered via stereotactic intracranial injection into the brain of C57BL6 mice, the recombinant virus infected a variety of cells in the vicinity of the lateral ventricle and in the rostral migratory stream, including numerous Ki67-positive proliferative neuroblasts (Fig. 5b). To test whether the *Bcan-Ntrk1* rearrangement can be induced *in vivo* and result in tumour formation, we next injected a cohort of *p53^{f/f}* mice (N = 7) with Ad-BN. To concomitantly induce loss

of *p53*, we co-injected these mice with recombinant adenoviruses expressing the Cre recombinase (Ad-Cre). As control, we used mice injected with Ad-Cre alone. These animals were monitored for up to 100 days for signs of tumour development. While none of the control, Ad-Cre only, mice developed gliomas within this time frame, 4/7 Ad-Cre; Ad-BN-injected mice developed high-grade gliomas (Fig. 5c). Histologically, the tumours were similar to those generated in the orthotopic implantation studies, and were positive for the glioma markers GFAP and Olig2 (Fig. 5d). Importantly, every tumour analysed showed the presence of the *Bcan-Ntrk1* rearrangement (Fig. 5e), indicating that the formation of the *Bcan-Ntrk1* gene fusion is strongly selected for *in vivo*. RNA collected from all tumours showed clear evidence of *Bcan-Ntrk1* mRNA expression by RT–PCR (Fig. 5f). Genomic DNA extracted from these tumours showed recombination of the *p53-floxed* allele relative to non-tumour adjacent tissue, indicating

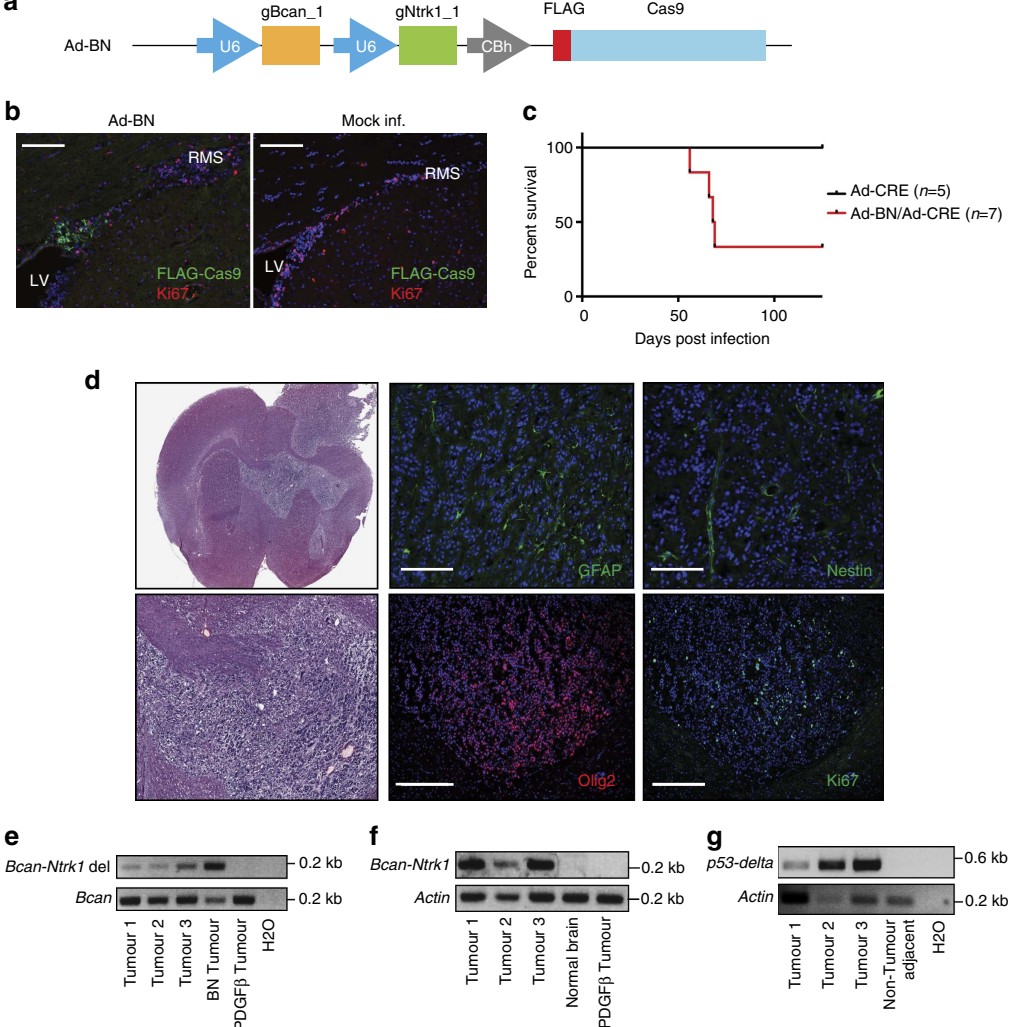

**Figure 5 | Generation of Bcan-Ntrk1-positive high-grade gliomas by *in vivo* somatic chromosomal engineering. (a)** Schematic of the recombinant adenoviral vector (Ad-BN) expressing the BN1 gRNA pair and FLAG-Cas9. (**b**) Immunostaining with anti FLAG and anti-Ki67 antibodies of brain sections from adult wild-type mice 48 h after stereotactical intracranial injection with Ad-BN (scale bar: 0.1 mm; LV = lateral ventricle; RMS = rostral migratory stream). (**c**) Survival curves of *p53^{f/f}* mice injected intracranially with a 1:1 mixture of Ad-BN and Ad-CRE (n = 7) or with Ad-CRE alone (n = 4). (**d**) H&E staining and immunostaining with the indicated antibodies of a representative tumour observed in mice infected with the Ad-BN virus (scale bar: 0.5 mm). (**e**) PCR analysis of genomic DNA purified from three representative gliomas observed in the Ad-BN/Ad-Cre-infected mice (Tumour 1–3), a *Bcan-Ntrk1-positive* glioma generated by orthotopic implantation of aNSC harbouring the *Bcan-Ntrk1* rearrangement (BN Tumour) and a PDGFb-driven mouse glioma were included as controls. Primers to detect the *Bcan-Ntrk1* deletion or the wild type Bcan allele were used. (**f**) RT–PCR using primers designed to detect the *Bcan-Ntrk1* fusion transcript was performed on total RNA extracted from the indicated tumours. (**g**) Detection of the recombined *p53* allele by genomic PCR on the indicated tumours and control tissues.

Ad-CRE-mediated loss of *p53* (Fig. 5g). These results conclusively demonstrate that *Bcan-Ntrk1* is a *bona fide* glioma driver and show that *in vivo* somatic chromosomal engineering can be used to model oncogenic chromosomal rearrangements in the mouse brain.

## Discussion

In this study, we describe an experimental pipeline that takes advantage of the flexibility of genome-editing technologies to rapidly interrogate the oncogenic potential of chromosomal rearrangements identified in human brain cancers. We applied this strategy to model four relatively rare chromosomal rearrangements of unclear oncogenic potential detected in human gliomas. We show that one of them, an intrachromosomal deletion generating a fusion between *BCAN* and *NTRK1*, drives the formation of aggressive high-grade gliomas and confers sensitivity to entrectinib, an experimental pan-Trk kinase inhibitor currently in clinical testing in a multicentre basket study (NCT02568267).

In contrast to laborious and costly gene targeting methods involving the modification of the mouse germline, the strategy we present here offers the important advantage of being easily scalable. Candidate rearrangements can be rapidly engineered in the relevant cellular context by cloning the appropriate gRNA pair into a Cas9-expressing plasmid[33], which is then nucleofected into aNSCs. aNSC clones harbouring the desired rearrangement can then be further characterized *in vitro*, orthotopically implanted into nude or syngeneic mice, or subjected to additional rounds of mutagenesis to model cooperating genetic events and to investigate mechanisms of acquired resistance to targeted therapies. We further show that these chromosomal rearrangements can be effectively induced directly in the brain of adult mice by stereotactic injection of recombinant adenoviruses expressing Cas9 and the two gRNAs. This approach can be used to conclusively demonstrate that a specific rearrangement can lead to glioma formation in a physiologic context and in an immunocompetent host.

Tumour formation driven by an *Ntrk* gene fusion has never before been modelled in the brain, although tumours driven by this important class of oncogene have been studied in other tissue contexts. Neurotrophin signalling drives cellular proliferation and survival through activation of Ras signalling[28]. *NF-1* mutation, which acts functionally to increase Ras activity, is a common driver mutation in mesenchymal glioblastoma[17]. It will be important for future studies to determine whether gliomas driven by *Ntrk* fusions also tend to fall into the mesenchymal subtype, which has a particularly poor prognosis. Although glioma patients are not routinely tested for *NTRK1* rearrangements, if ongoing clinical trials with entrectinib confirm the encouraging survival data obtained in our preclinical model, it will be important to develop simple assays, based on RT–PCR or interphase FISH, to identify *NTRK1* fusions in brain tumours.

While induction of the *Bcan-Ntrk1* rearrangement readily resulted in high-grade gliomas, we never observed tumours after implantation of *Fgfr3-Tacc3*, *Sec61g-Egfr* or *Gga2-Prkcb* positive cell populations within the timeframe of analysis (150 days). This result is especially surprising for the *Fgfr3-Tacc3* rearrangement, because a previous study had shown that lentiviral transduction of the *Fgfr3-Tacc3* cDNA in the mouse cortex in combination with *p53* knockdown can induce the formation of gliomas[21]. Although we did not observe glioma formation even after inducing the *Fgfr3-Tacc3* rearrangement directly in the brain of *p53^f/f^* mice using recombinant adenoviruses (Ad-FT, Supplementary Fig. 8), it is formally possible that the failure to induce *Fgfr3-Tacc3*-positive tumours reflects the inability of our adenoviral vector to infect the relevant cell of origin. Another plausible explanation for this discrepancy is that cDNA expression from the artificial viral promoter in the previous study results in higher expression levels than those associated with the actual rearrangement. Consistent with this hypothesis is a recent study showing that *FGFR3-TACC3*-positive human glioblastomas consistently display focal amplification of the rearranged locus[14]. The *EGFR* locus is similarly amplified with high frequency in human patients[14], and insufficient expression from the non-amplified endogenous *Sec61g* promoter may also account for our failure to observe *Sec61g-Egfr*-driven tumours. The *GGA2-PRKCB* fusion differs from the others examined in this study in that it has been found exclusively in low-grade glioma, which has a distinct mutational signature from high-grade glioblastoma. Specifically, the *IDH1 R132H* neomorphic allele is found at high frequency in grade II–III glioma[34] and may be an important cooperating mutation with potentially transforming oncogenes such as *GGA2-PRKCB*. Future studies using CRISPR technology to introduce additional mutations into clonal *p53^-/-^* aNSC cell lines bearing these rearrangements may shed light on the specific combinations of mutations required for full transformation and brain tumour formation.

We predict that variations of this general strategy will be useful to model novel gene fusions associated with other tumour types, allowing the functional characterization of the rapidly growing number of gene fusions identified in human cancers. We also predict that this work will greatly facilitate the study of chromosomal rearrangements that do not result in the formation of gene fusions, but that could promote tumorigenesis by interfering with gene regulatory elements such as enhancers or insulators, or affect gene expression by altering high-order chromatin structure[35].

## Methods

**Orthotopic cell implantation.** Primary adherent mouse aNSC or glioma cells were collected using Accutase (Sigma) and resuspended in phosphate-buffered saline at a concentration of 166,000 cells $\mu l^{-1}$. Four- to six-week-old female nude mice (NCr-*Foxn1^nu^*) were anaesthetized using ketamine/xylazine and administered 3 $\mu l$ (500,000 cells) of this cell suspension via stereotactic intracranial injection at 1 mm $-1$ mm$^{-1}$ from bregma at a depth of 2 mm. Mice were monitored for 150 days or until they became symptomatic (weight loss, hunched posture, neurological symptoms), at which point they were killed and subjected to full necropsy.

**Animal models.** C57BL/6J wild type, *Trp53^flox/flox^* and *Trp53^-/-^* mice in C57BL/6J genetic background were acquired from The Jackson Laboratory and have been previously described[36,37]. Athymic nude mice (NCr-*Foxn1^nu^*) were acquired from Taconic. Purified, high titre adenovirus for Ad-BN, Ad-FT and Ad-CRE viral constructs was produced by Viraquest. A mix of male and female *Trp53^flox/flox^* or C57BL/6 wild type mice were administered $\sim 3 \times 10^9$ infectious particles via stereotactic intracranial injection at 1 mm $-1$ mm$^{-1}$ from bregma at a depth of 2 mm. For *in vivo* experiments using the Trk inhibitor entrectinib (Ignyta), tumour-bearing athymic nude females were administered 50 mg kg$^{-1}$ entrectinib or vehicle control (0.5% methyl cellulose) via daily oral gavage for 14 days. Animals were randomly assigned to control or experimental groups after implantation with tumour cells. A total of 10 mice per group was selected based on previous studies[29]. Investigators were not blind with respect to treatment. Brain magnetic resonance imagings were acquired on a Bruker 4.7T Biospec scanner by the Animal Imaging Core Facility at Memorial Sloan Kettering Cancer Center.

All mouse experiments were approved by MSKCC's Institutional Animal Care and Use Committee. Tumour histopathology was reviewed by a board-certified pathologist that specializes in human primary brain tumours (M.K.R.).

**Immunostaining.** Whole brains were fixed in 4% paraformaldehyde and dehydrated in sucrose before embedding in OCT (Sakura) and sectioning (10 $\mu m$) using a Leica Cryostat. Antigen retrieval was performed by boiling slides in 10 mM sodium citrate buffer, followed by blocking/permeabilization in phosphate-buffered saline, 10% normal goat serum, 0.5% TritonX-100. Primary antibody incubation was followed by incubation with an appropriate fluorescent secondary antibody (Alexa 488 or Alexa 568 anti-Rabbit or anti-Mouse (Invitrogen, 1:1,000)) for 1 h and counterstaining with 5 $\mu g\, ml^{-1}$ DAPI (Invitrogen). The following primary

antibodies were used: Ki67 (Cell Signaling Technology #9449S, 1:1,000), GFAP (Millipore, #MAB3402, 1:500), Olig2 (Millipore, #AB9610, 1:500), CD105 (AbCam, #AB2529, 1:200), Nestin (BD Biosciences, #556309, 1:500). Images were acquired with a Zeiss Axioplan2 Imaging Widefield Microscope using the AxioVision software (Zeiss). pAKT-S473 stainings (Cell Signaling Technology, #9271S, 1:1,000) were performed on formalin-fixed paraffin-embedded tumour slices by the MSKCC Molecular Cytology Core facility. TUNEL staining is described in Supplementary Methods. Images were processed with Adobe Photoshop and assembled with an Adobe Illustrator. Image quantification was performed using ImageJ software from >3 representative images from each sample.

**Cell culture.** For primary aNSC cultures, 4- to 6-week-old $p53^{-/-}$ mice were killed and the subventricular zone of each lateral ventricle was microdissected, pooled and digested using papain (Worthington Biochemical). The digestion was neutralized using ovomucoid (Worthington Biochemical) and the resulting cell suspension was plated under non-adherent conditions in Neurocult Stem Cell Basal Media with Proliferation Supplements, 20 ng ml$^{-1}$ EGF, 10 ng ml$^{-1}$ FGF and 2 µg ml$^{-1}$ heparin (Stem Cell Technologies) to allow for the outgrowth of NSC-derived neurospheres. After 8–10 days, spheres were collected, disassociated using accutase (Sigma) and seeded into tissue culture plates pre-coated with 10 µg ml$^{-1}$ Laminin (Sigma). Tumours were digested using papain (Worthington Biochemical), triturated and neutralized with ovomucoid (Worthington Biochemical) according to the manufacturer's instructions. Resulting cells were seeded into tissue culture plates pre-coated with 10 µg ml$^{-1}$ Laminin (Sigma) in Neurocult Stem Cell Basal Media with Proliferation Supplements, 20 ng ml$^{-1}$ EGF, 10 ng ml$^{-1}$ FGF and 2 µg ml$^{-1}$ heparin (Stem Cell Technologies).

NIH3T3 mouse fibroblasts were acquired from ATCC and maintained in Dulbecco's modified Eagle's media (ATCC) supplemented with 10% fetal bovine serum (ATCC). Proliferating NIH3T3 cells were infected with Ad-FT at $2 \times 10^9$, $2 \times 10^{10}$, $2 \times 10^{11}$ and $2 \times 10^{12}$ infectious particles per million cells and collected after 48 h for genomic DNA analysis.

**Cell viability assays.** Adherent mouse glioma primary cultures were seeded into laminin-coated 48-well plates at a density of 10,000 cells well$^{-1}$. Twenty-four hours after seeding, cells were incubated in growth factor-free media. Twenty-four hours after growth factor withdrawal, media was again changed and triplicate wells were treated with dimethyl sulphoxide or increasing concentrations of entrectinib in the absence of growth factor. Twenty-four hours after initiation of treatment, media was replaced with matched treatment media supplemented with 10% Alamar Blue (Invitrogen). Absorbance (A570/A600) was read for each well at 6 and 24 h.

**Western blotting.** Fifteen micrograms of protein were separated on 10% acrylamide/bisacrylamide gels, transferred onto PVDF membranes and blocked for 30 min–1 h in 10% fat-free milk in TBS-T. The following primary antibodies were used: β-actin (Sigma, A2066, 1:5,000), β-tubulin (Sigma, # T4026, 1:5,000), p-ERK1/2 (Cell Signaling Technology, # 9101S, 1:1,000), ERK1/2 (Cell Signaling Technology, # 4659P, 1:1,000), p-Akt-S473 (Cell Signaling Technology, # 4060S, 1:1,000), Akt (Cell Signaling Technology, # 9272S, 1:1,000), cleaved Caspase-3 (Cell signaling Technology, #9661S, 1:1,000). Uncropped immunoblot images can be found in Supplementary Fig. 9.

**Plasmids and adenoviral vectors.** The pX330 vector-expressing Cas9 (Addgene plasmid 42230) was digested with *Bbs*I and ligated to annealed and phosphorylated gRNA oligonucleotides targeting *Bcan*, *Ntrk1*, *Fgfr3*, *Tacc3*, *Sec61g*, *Egfr*, *Gga2* and *Prkcb*. The genomic target sequences for each sgRNA used in this study are listed in Supplementary Table 2. For cloning of tandem U6-gRNA-Cas9 constructs, the second U6-gRNA cassette was amplified using primers containing the *Xba*I and *Kpn*I sites and cloned into the pX330 construct containing the appropriate gRNA. For Adeno-Bcan-Ntrk1 cloning, pX330-Bcan-Ntrk1 vector was modified by adding an *Xho*I site upstream the first U6 promoter. An *Eco*RI/*Xho*I fragment containing the double U6-gRNA cassette and the Flag-tagged Cas9 was then ligated into the *Eco*RI/*Xho*I-digested pacAd5 shuttle vector. A similar method was used to generate the Adeno-Fgfr3-Tacc3 vector.

**Plasmid nucleofection.** Nucleofection was performed using the mouse NSC nucleofector kit (Lonza) according to the manufacturer's instructions. Briefly, adherent growing $p53^{-/-}$ mouse aNSCs were collected using accutase and ~1 × 10$^6$ cells were mixed with 1 µg each expression plasmid for a total of 2 µg per nucleofection. Electroporations were performed using an Amaxa Nucleofector device, and nucleofection efficiency was determined by FACS analysis of GFP-expressing cells 48 h post-nucleofection.

**PCR and RT–PCR analysis.** For PCR analysis of genomic DNA, cells or tissue were collected and genomic DNA was collected using a DNeasy Blood and Tissue kit (Qiagen) according to the manufacturer's instructions. For RT–PCR, total RNA was purified from cells or tissue using an RNeasy mini kit (Qiagen) according to

the manufacturer's instructions. cDNA was generated using the Superscript IV kit (Invitrogen). The primers used in the various PCR reactions are provided in Supplementary Table 3.

**Statistics.** Unpaired *t*-test and one-way ANOVA were performed on relevant data sets using GraphPad Prism 6 software. Where appropriate, Tukey's test was used for *post hoc* analysis of ANOVA. For Kaplan–Meier survival curve data, log-rank (Mantel Cox) test was performed using GraphPad Prism 6. For all analyses, *P*-values <0.05 were considered significant. Data are shown as mean ± s.e.m.

**Data availability.** All data supporting the findings of this study are available within the article and Supplementary Files, or available from the authors upon request.

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

## Acknowledgements

We would like to thank Jason Huse for technical assistance and advise regarding glioma tumour histology. We thank Riddhi Shah and Paul Ogrodowski for animal husbandry assistance. This work was funded in part by grants from the Pershing Square-Sohn Cancer Research Foundation (to A.V.), the MSKCC Brain Tumor Center (to A.V. and R.B.), the Geoffrey Beene Cancer Research Foundation (to A.V.) and Cancer Center Support Grant P30 CA008748 (to M.S.). R.K. is funded by the T32 CA 160001 MSKCC NIH TROT fellowship. This work was supported by grant I10-0095 from the STARR foundation.

## Author contributions

P.J.C., R.B. and A.V. conceived the project, designed the experiments and wrote the manuscript. P.J.C. performed the experiments on aNSCs, the orthotopic implantations and the intracranial viral injections. P.J.C. and R.K. generated and tested the constructs used to produce recombinant adenoviruses. R.T. administered entrectinib to tumour-bearing mice, collected data and analysed results of these experiments. E.S.d.L. performed western blots, PCR assays and immunostaining experiments. A.D. and M.S. assisted in designing and interpreting experiments to assess entrectinib efficacy. The histopathology was analysed and reviewed by M.K.R.

## Additional information

**Competing interests:** A.D. has received honoraria from Ignyta. The remaining authors declare no competing financial interests.

DOI: 10.1038/ncomms16187   OPEN

# Author Correction: Somatic chromosomal engineering identifies BCAN-NTRK1 as a potent glioma driver and therapeutic target

Peter J. Cook, Rozario Thomas, Ram Kannan, Esther Sanchez de Leon, Alexander Drilon, Marc K. Rosenblum, Maurizio Scaltriti, Robert Benezra & Andrea Ventura

Nature Communications 8:15987 doi: 10.1038/ncomms15987 (2017); Published 11 Jul 2017; Updated 13 Mar 2018

In the original version of this Article, financial support was not fully acknowledged. The PDF and HTML versions of the Article have now been corrected to include the following:

'This work was supported by grant I10-0095 from the STARR foundation.'

