## [Peer Review File · Nature Communications]

REVIEWERS' COMMENTS:

Reviewer #1 (Remarks to the Author):

In this manuscript, Ventura and colleagues describe an approach to creating fusion alleles identified by genome characterization efforts and testing their function in genetically engineered mouse models. In particular, they show that creation of a fusion between Bcan-Ntrk1 in a p53 null background leads to cells that form orthotopic tumors that response to a NTRK1 inhibitor and that engineering this allele in adult mice leads to glioblastomas.

The studies provide strong preclinical evidence that this fusion event is oncogenic and a potential target in the 1-2.5% of glioblastoma patients who harbor this allele. Although less novel, these studies also demonstrate the utility of performing CRISPR-Cas9 editing to more rapidly generate models of specific fusion events. Given the strong evidence that this provides for testing a NTRK inhibitor in a selected patient population, these studies are timely and important.

Although there is some evidence that this fusion is oncogenic, these studies provide strong in vivo support for consideration of testing agents that target these fusions clinically.

Reviewer #2 (Remarks to the Author):

This well-executed study by Cook et al. utilizes genome engineering to generate new glioblastoma models based on frequently observed genomic re-arrangements that lead to gene fusions. This is an elegant and important study that will have broader implications for using CRISPR/Cas9 genome engineering to model rearrangements/fusions in adult, pediatric gliomas and other cancer types. The study provides a robust pipeline to validate fusion events in vitro and in vivo, followed by therapeutic pre-clinical approaches.

The authors have addressed most of the major concerns regarding cell of origin and the inability of other fusions to give rise to tumors either with additional experiments or by elaborating in the discussion. I have no further comments or concerns.

Response to reviewers

REVIEWERS' COMMENTS:

Reviewer #1 (Remarks to the Author):

In this manuscript, Ventura and colleagues describe an approach to creating fusion alleles identified by genome characterization efforts and testing their function in genetically engineered mouse models. In particular, they show that creation of a fusion between Bcan-Ntrk1 in a p53 null background leads to cells that form orthotopic tumors that response to a NTRK1 inhibitor and that engineering this allele in adult mice leads to glioblastomas.

The studies provide strong preclinical evidence that this fusion event is oncogenic and a potential target in the 1-2.5% of glioblastoma patients who harbor this allele. Although less novel, these studies also demonstrate the utility of performing CRISPR-Cas9 editing to more rapidly generate models of specific fusion events. Given the strong evidence that this provides for testing a NTRK inhibitor in a selected patient population, these studies are timely and important.

Although there is some evidence that this fusion is oncogenic, these studies provide strong in vivo support for consideration of testing agents that target these fusions clinically.

Reviewer #2 (Remarks to the Author):

This well-executed study by Cook et al. utilizes genome engineering to generate new glioblastoma models based on frequently observed genomic re-arrangements that lead to gene fusions. This is an elegant and important study that will have broader implications for using CRISPR/Cas9 genome engineering to model rearrangements/fusions in adult, pediatric gliomas and other cancer types. The study provides a robust pipeline to validate fusion events in vitro and in vivo, followed by therapeutic pre-clinical approaches.

The authors have addressed most of the major concerns regarding cell of origin and the inability of other fusions to give rise to tumors either with additional experiments or by elaborating in the discussion. I have no further comments or concerns.

Response:

We thank these reviewers for the constructive criticisms provided during the previous rounds of review and we are glad that they are satisfied by our efforts to address them. We believe that the paper has been greatly improved in the process.